# Molecular mechanism of antihistamines recognition and regulation of the histamine H$_1$ receptor

Dandan Wang[1,3], Qiong Guo[1,3], Zhangsong Wu[2,3], Ming Li[1], Binbin He[1], Yang Du [2], Kaiming Zhang [1] ✉ & Yuyong Tao [1] ✉

Histamine receptors are a group of G protein-coupled receptors (GPCRs) that play important roles in various physiological and pathophysiological conditions. Antihistamines that target the histamine H$_1$ receptor (H$_1$R) have been widely used to relieve the symptoms of allergy and inflammation. Here, to uncover the details of the regulation of H$_1$R by the known second-generation antihistamines, thereby providing clues for the rational design of newer antihistamines, we determine the cryo-EM structure of H$_1$R in the apo form and bound to different antihistamines. In addition to the deep hydrophobic cavity, we identify a secondary ligand-binding site in H$_1$R, which potentially may support the introduction of new derivative groups to generate newer antihistamines. Furthermore, these structures show that antihistamines exert inverse regulation by utilizing a shared phenyl group that inserts into the deep cavity and block the movement of the toggle switch residue W428$^{6.48}$. Together, these results enrich our understanding of GPCR modulation and facilitate the structure-based design of novel antihistamines.

Histamine is a biogenic amine that plays an important role in various physiological and pathophysiological conditions by activating four different G-protein coupled receptors (GPCRs), classified as H$_1$R, H$_2$R, H$_3$R, and H$_4$R[1–3]. These four receptors are either well-established drug targets (H$_1$R, H$_2$R, and H$_3$R) or are being evaluated for related diseases, for example, by targeting H$_4$R for anti-inflammatory diseases[4–6]. Among them, H$_1$R is widely distributed throughout the body in epithelial cells and smooth vascular, neuronal, glial, and immune cells[7,8]. When activated by histamine released by mast cells or basophils, H$_1$R can cause allergic and inflammatory symptoms and has therefore been extensively targeted in the development of antihistamines[9,10]. To date, more than 45 H$_1$R-antihistamines are available worldwide, and moreover, new antihistamines are continuously being investigated[11,12]. Since the first antihistamine came out in 1937, antihistamines have gone through the development process of first-generation, second-generation and now the new second-generation antihistamines[5,13,14]. Although first-generation antihistamines such as chlorpheniramine and diphenhydramine were once widely used clinically, they cause adverse central nervous system (CNS) responses due to their poor receptor selectivity and ability to cross the blood-brain barrier (BBB) and bind to H$_1$R in the CNS[11,15–17]. By introducing new chemical groups, such as carboxyl groups combined with protonated amines, second-generation antihistamines have been successively developed. Second-generation antihistamines have high H$_1$R selectivity, rarely cross the BBB, and preferentially bind peripheral H$_1$R[11,18]. Therefore, second-generation antihistamines cause few or no CNS side effects[19,20]. However, second-generation H$_1$R antihistamines may still have adverse effects on the heart[21]. For example, the second-generation drug

[1]Department of Laboratory Medicine, The First Affiliated Hospital of USTC, MOE Key Laboratory for Membraneless Organelles and Cellular Dynamics, Hefei National Center for Cross-disciplinary Sciences, Biomedical Sciences and Health Laboratory of Anhui Province, Center for Advanced Interdisciplinary Science and Biomedicine of IHM, Division of Life Sciences and Medicine, University of Science and Technology of China, 230027 Hefei, P. R. China. [2]Kobilka Institute of Innovative Drug Discovery, School of Medicine, The Chinese University of Hong Kong, 518172 Shenzhen, Guangdong, China. [3]These authors contributed equally: Dandan Wang, Qiong Guo, Zhangsong Wu. ✉e-mail: kmzhang@ustc.edu.cn; taoyy@ustc.edu.cn

astemizole was reported to cross-interact with several other targets and thus cause cardiotoxicity in vivo[22].

Recently, new second-generation antihistamines, mainly derived from the active metabolites or optical isomers of second-generation antihistamines, have been introduced[23]. For example, as a metabolite of the second-generation antihistamine loratadine, desloratadine has been developed and defined as a new-generation antihistamine[24,25]; however, it remains doubtful whether these newer antihistamines are actually superior to their predecessors in terms of efficacy and safety. In other words, great challenges still exist in the development of new antihistamines, especially the lack of interaction details between $H_1R$ and second-generation antihistamines, which hinders the rational design of new drugs. In addition to guiding drug development, elucidating the structural states of $H_1R$ will also help to understand the mechanism of inverse modulation of ligands on GPCRs. Since $H_1R$ has a high basal signaling capacity, a prominent feature of many GPCRs, and all antihistamines have been shown to be inverse agonists rather than antagonists[26–28], how $H_1R$ obtains its constitutive activity and how it is blocked by antihistamines remain poorly elucidated. Here, we determine the structure of $H_1R$ in the apo form and bound to different antihistamines. The structural information provides insights into the interaction and modulation of $H_1R$ with antihistamines and will facilitate the structure-based design of next-generation drugs.

## Results

### Structure determination of $H_1R$ in apo form and bound to inverse agonists

To solve the structure of $H_1R$ in complex with inverse agonists, we employed the mBril fusion and gluing strategy recently developed by our group[29]. Briefly, mBril was fused between TM5 and TM6 in a way that would form two continuous helices with the receptor (Supplementary Fig. 1a). Meanwhile, the helical tag K3-ALFA was added to the C-terminus of H8 in the desired configuration (Supplementary Fig. 1a). We next used Alphafold2 to predict the structure of the fusion protein. By docking the known 1B3-Bril structure onto the predicted receptor, we obtained a complex model, and based on this model, the "4-9" glue molecule was chosen for the first attempt to solve the structure of $H_1R$ bound to the first-generation antihistamine mepyramine. However, this sample did not show many particles with the desired shape in the 2D analysis (Supplementary Fig. 1b). The reason seems to be that this "4-9" glue molecule is too harsh since the mBril domain is bent in most of the particles (Supplementary Fig. 1b). We then reprepared the grid by using the "6-13" glue molecule. This new sample indeed displayed much better particles in the preanalysis (Supplementary Fig. 1c), and we thus conducted large-scale data collection. After data processing, a cryo-EM map with a global resolution of 3.2 Å was obtained (Supplementary Fig. 2 and Supplementary Table 1). A further refinement focused on the TM region produced an improved map in which the side chains of most $H_1R$ residues were traceable (Fig. 1a, Supplementary Fig. 2). Guided by the map, mepyramine and $H_1R$ residues consisting of S24-Q171, V174-L221 and N408-R481 were built.

The same structure determination strategy was then applied to $H_1R$ in apo form and complexed with the second-generation antihistamines astemizole and desloratadine, and their structures were finally determined at 3.5, 3.0 and 3.4 Å resolution, respectively (Fig. 1b–d, Supplementary Figs. 3–5 and Supplementary Table 1). Interestingly, although following exactly the same conditions, the apo form of $H_1R$ had lower quality density maps compared to those complexed with inverse agonists (Fig. 1 and Supplementary Figs. 2–5), suggesting that without inverse agonist stabilization, $H_1R$ may have a dynamic structure (Supplementary Fig. 6). Guided by the density map, models of $H_1R$ in apo form and bound to astemizole and desloratadine were built (Fig. 1b–d). Finally, the model of $H_1R$ in apo form contains residues P29-E55, L58-R134, R139-Q170, R176-L221 and N408-R481. The astemizole-$H_1R$ model contains residues T20-Q171, V174-L221, and

N408-R481. The desloratadine-$H_1R$ model contains residues P25-Q171, D178-L221, and N408-R481. As expected, all the inverse agonists were bound at the orthosteric pocket with the core pharmacophore buried at roughly similar positions (Fig. 1a−c). $H_1R$ shares an almost identical overall conformation in the different inverse agonist bound states, including that in complex with doxepin[30], as reflected by the root-mean-square deviation (RMSD) values of 0.5-0.7 Å for the Cα within the receptor. Although all these $H_1R$ structures reside in a typical inactive conformation, featuring the close arrangement of TM6 with the receptor core on the intracellular side (Fig. 1), noticeable structural differences are also observed around the ligand binding pocket, as discussed below.

### Recognition mechanism of mepyramine with $H_1R$

Mepyramine is a first-generation antihistamine that targets $H_1R$[31,32] and has been approved for the treatment of allergic reactions and urticaria. Like many other first-generation antihistamines, mepyramine can cross the blood–brain barrier and often causes serious side effects, such as drowsiness. Recently, mepyramine has been shown to bind to a variety of voltage-gated sodium channels and directly inhibit their activity, so mepyramine has the potential to be developed as a topical analgesic agent[33]. Deciphering the recognition details of mepyramine by each individual target receptor will provide valuable clues for further optimization of the drug for improved specificity and efficacy.

The determined structure of mepyramine-$H_1R$ shows that mepyramine is captured in the orthosteric pocket mainly constituted by residues from TM3, TM5, TM6, and TM7 (Fig. 2a). The three functionality groups of mepyramine establish an interaction network with $H_1R$. Among them, the methoxyphenyl group of mepyramine, especially the methoxy moiety, is deeply inserted into a hydrophobic cavity formed by $H_1R$ residues T112$^{3.37}$, I115$^{3.40}$, N198$^{5.46}$, F199$^{5.47}$, F424$^{6.44}$, W428$^{6.48}$ and F432$^{6.52}$ (Fig. 2a). Consistent with the structural observation, all the single mutations of F199$^{5.47}$, W428$^{6.48}$ and F432$^{6.52}$ abolished the $H_1R$ affinity with mepyramine[34]. In contrast, the N198A mutation did not cause obvious affinity loss[35], suggesting that hydrophobic contacts dominate the $H_1R$ interaction with the methoxyphenyl group. Notably, W428$^{6.48}$ is the toggle switch residue that normally initiates GPCR activation through ligand-induced conformational changes[36]. In addition, F424$^{6.44}$ and I115$^{3.40}$ are derived from the P$^{5.50}$-I$^{3.40}$-F$^{6.44}$ triadmotif, which also plays a key role in GPCR activation[36]. In general, GPCR ligands bind receptors at sites above these motifs, as is the case for $H_1R$ recognition of the natural ligand histamine (Supplementary Fig. 7b); however, when coordinating an inverse agonist, $H_1R$ adopts a completely different binding mode, and to the best of our knowledge, the site holding mepyramine is the deepest location ever observed. The pyridine moiety of mepyramine is again mainly coordinated by hydrophobic interactions with $H_1R$ residues Y108$^{3.33}$, W158$^{4.56}$, A195$^{5.43}$, F432$^{6.52}$ and F435$^{6.55}$ (Fig. 2a). Among them, as anticipated, the single mutations of Y108$^{3.33}$ and W158$^{4.56}$ completely disrupted the mepyramine binding activity[34]. The F435A mutation (F436A in guinea pig $H_1R$) also resulted in a 4-fold and 20-fold decrease in the affinity of mepyramine for human and guinea pig $H_1R$[35,37], respectively. The third functionality group, the dimethylamino moiety, adopts a pose similar to that in the doxepin-$H_1R$ complex[30] and forms a salt bridge with the $H_1R$ residue D107$^{3.32}$ (Fig. 2a and Supplementary Fig. 7b). D107$^{3.32}$ is a highly conserved residue in aminergic receptors and has been repeatedly confirmed to be indispensable for ligand binding[30,38]; therefore, D107$^{3.32}$-mediated polar contact may represent a universal mode in $H_1R$ ligand interactions. Moreover, Y431$^{6.51}$ forms a hydrogen bond with the N atom in the dimethylamino group (Fig. 2a). The single mutant of Y431$^{6.51}$ also completely disrupted the binding activity of mepyramine[34]. In summary, the structural and functional data reveal that $H_1R$ engages the first-generation drug mepyramine through a highly hydrophobic pocket and a conserved salt bridge.

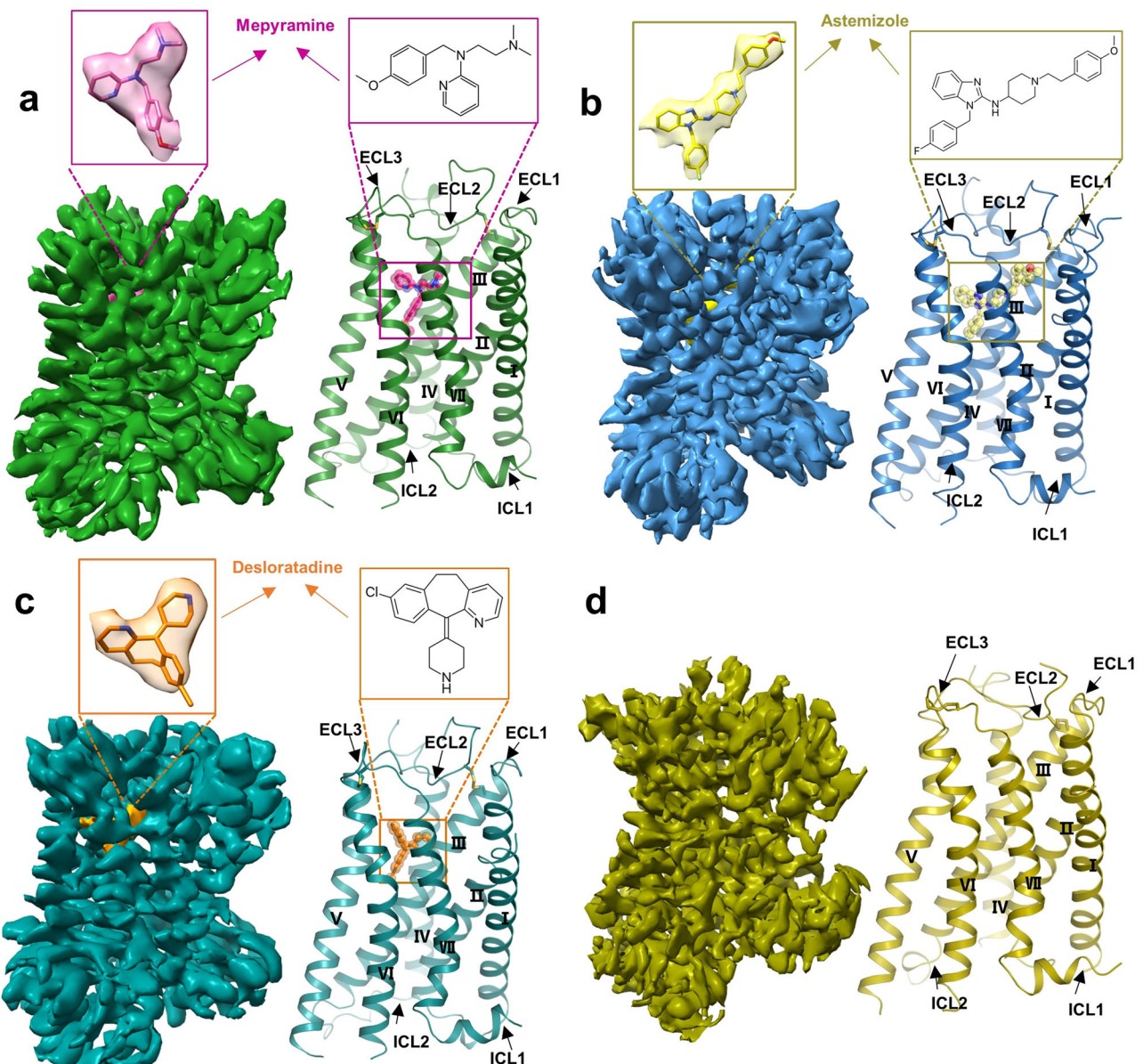

**Fig. 1 | Cryo-EM map and overall structure of H₁R bound to inverse agonist and H₁R in apo form. a** Cryo-EM map and overall structure of the H₁R-mepyramine complex. H₁R and mepyramine are colored green and hot pink, respectively. **b** Cryo-EM map and overall structure of the H₁R-astemizole complex. H₁R and astemizole are colored sky blue and pale yellow, respectively. **c** Cryo-EM map and overall structure of the H₁R-desloratadine complex. H₁R and desloratadine are colored in deep teal and orange, respectively. **d** Cryo-EM map and overall structure of H₁R in apo form, H₁R color as olive.

## Recognition mechanism of astemizole with H₁R

Astemizole is a second-generation antihistamine drug, but it was withdrawn from the market due to its potential to cause arrhythmias at high doses. The reason is that astemizole not only targets H₁R but also blocks the hERG potassium channel[39,40]. In addition, astemizole has also been shown to disrupt the protein–protein interaction within polycomb repressive complex 2 (PRC2) and thus arrest the proliferation of cancer cells[41]. Repurposing astemizole for new clinical uses has been constantly explored[42–44]. Delineating the molecular recognition mechanism with astemizole is a prerequisite for precise modification of astemizole to eliminate its pleiotropic effect.

The H₁R structure bound to astemizole shows that the fluorophenyl and benzimidazole groups of astemizole occupy positions similar to those accommodating the methoxyphenyl and pyridine groups of mepyramine, respectively (Fig. 2b). A hydrogen bond, also mediated by Y431$^{6.51}$, and the remaining extensive hydrophobic interactions are responsible for the coordination of astemizole (Fig. 2b). The piperidine group extends as the dimethylamino moiety of mepyramine and establishes an electrostatic interaction with the H₁R residue D107$^{3.32}$ (Fig. 2b). The shared pose of mepyramine and astemizole in the orthosteric pocket suggests that H₁R adopts a conserved binding mode for ligands with similar chemotypes. Interestingly, the additional methoxyphenyl moiety of astemizole is extended from the main ligand binding pocket toward the site defined by residues from ECL2, TM2, TM3 and TM7 (Fig. 2b). We refer to this site as a secondary binding pocket. Surrounding H₁R residues Y87$^{2.64}$, W103$^{3.28}$, and M451$^{7.36}$ make hydrophobic contacts with the methoxyphenyl moiety (Fig. 2b). Mutation of Y87$^{2.64}$ and W103$^{3.28}$ markedly impaired the ability of astemizole to inhibit H₁R signaling (Fig. 2c), confirming the contribution of this secondary pocket to astemizole binding. Apart from the hydrophobic residues, three polar residues, N84$^{2.61}$, K179$^{ECL2}$ and H450$^{7.35}$, also decorate this secondary pocket (Fig. 2b); however,

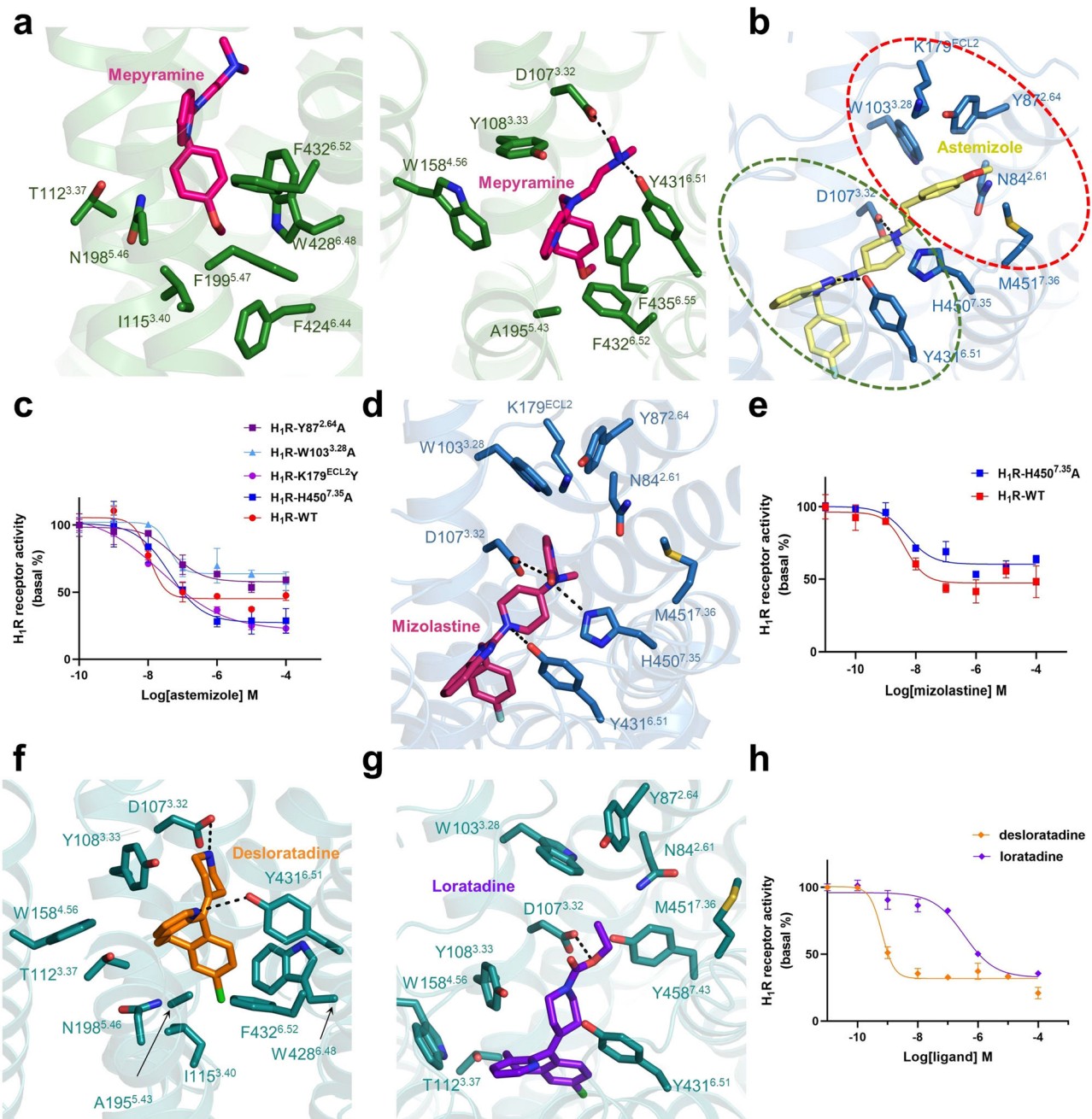

**Fig. 2 | Detailed interactions of inverse agonists in the H₁R ligand-binding pocket. a** The ligand-binding pocket of mepyramine. H₁R is shown as green ribbons, with critical residues for ligand binding shown as green sticks and mepyramine shown as hot pink sticks. **b** The ligand-binding pocket of astemizole. H₁R and astemizole are sky blue and pale yellow, respectively. The main and secondary pockets are indicated with green and red circles, respectively. **c** Dose-dependent responses of astemizole measured by cellular IP1 accumulation assays in wild-type and mutant H₁R. **d** Mizolastine docked in H₁R. Mizolastine is shown as warm pink sticks, and the critical residues of H₁R are shown as blue sticks. The H450^7.35-mediated hydrogen bond is indicated with a dashed line. **e** Dose-dependent

responses of mizolastine measured by cellular IP1 accumulation assays in wild-type and mutant H₁R. **f** The ligand-binding pocket of desloratadine. H₁R and desloratadine color as deep teal and orange, respectively. **g** Loratadine docked in H₁R. Loratadine and H₁R are colored purple–blue and deep teal, respectively. Hydrogen bonds and salt bridges are highlighted as black dashed lines. **h** Dose-dependent responses of desloratadine (orange line) and loratadine (purple line) and loratadine at wild-type H₁R measured by cellular IP1 accumulation assays. The data from the cellular IP1 accumulation assays are represented as the mean ± SEM, n = 3 independent samples.

they do not form any polar interactions with the methoxyphenyl moiety and even induce forces inconsistent with the hydrophobicity of the methoxyphenyl moiety. Mutating K179^ECL2 and H450^7.35 to hydrophobic residues improves the efficacy of astemizole in inhibiting H₁R (Fig. 2c and Supplementary Table 2). Therefore, modifying the methoxyphenyl moiety with alternative derivative groups, for example, introducing hydrogen bond donors/acceptors, may further improve

the specificity of astemizole. In other words, making the most use of this secondary pocket could lead to new selective or effective antihistamines. Consistent with this notion, another second-generation nonsedating antihistamine, mizolastine, which differs from astemizole only by the methoxyphenyl moiety, does not cross-interact with any non-H₁R targets, including the potassium channel[45]. To probe how mizolastine engages H₁R, especially how the polar group

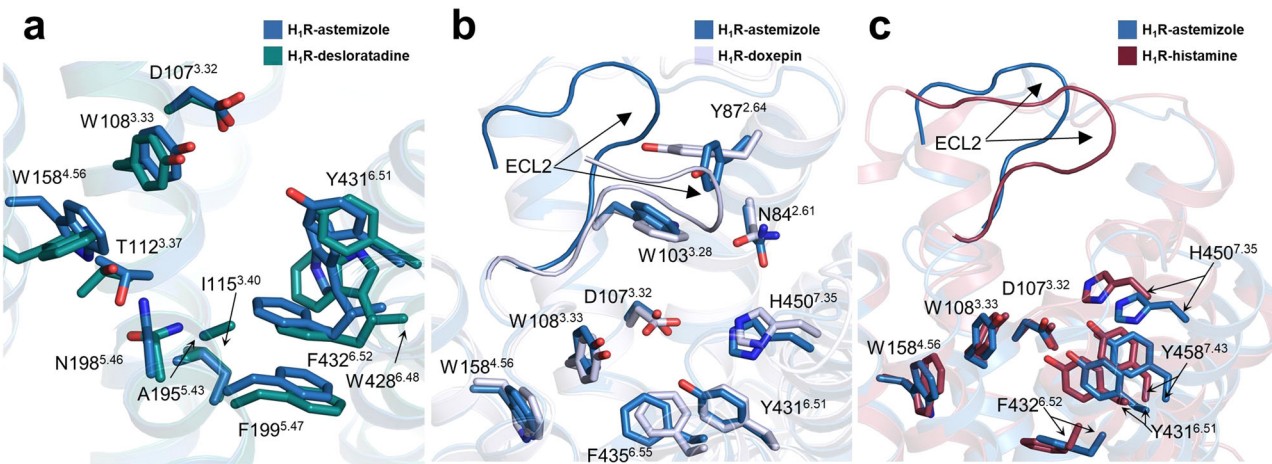

**Fig. 3 | The ligand-binding pocket of different H₁R complexes. a** Superposition of H₁R-astemizole (sky blue) and H₁R-desloratadine (deep teal). **b** Comparison of the selective ligand-binding pockets of H₁R-astemizole (sky blue) and H₁R-doxepin (light blue) (PDB ID: 3RZE). **c** Comparison of the crucial residues and ECL2 from H₁R-astemizole (sky blue) and H₁R-histamine (raspberry). Key different residues are shown as sticks.

dihydropyrimidine of mizolastine fits into the secondary pocket, we docked mizolastine onto H₁R (Supplementary Table 3). In the resulting docked structure, mizolastine occupies the expected position in the pocket, and its fluorophenyl and benzimidazole groups adopt the same poses as the corresponding groups in astemizole, indicating that the docking of mizolastine is plausible (Fig. 2d). Interestingly, instead of inducing an unfavorable environment for the methoxyphenyl moiety of astemizole, H450[7.35] mediates a hydrogen bond with the dihydropyrimidine of mizolastine (Fig. 2d), suggesting the compatibility of this derivative group with the secondary pocket. Accordingly, different from the boosting effect of astemizole, the H450A mutation obviously crippled the ability of mizolastine to inhibit H₁R signaling (Fig. 2e and Supplementary Table 2). Therefore, while the incorporation of dihydropyrimidine abolishes the mizolastine affinity for other non-H₁R targets, the dihydropyrimidine group can be well matched to the secondary pocket of H₁R, thereby still retaining a high H₁R affinity. Together, the astemizole-H₁R structure reveals that in addition to the main ligand binding pocket, H₁R possesses a secondary pocket that can be explored for the development of new antihistamines.

### Recognition mechanism of desloratadine with H₁R

Desloratadine, an antihistamine for the treatment of seasonal allergic rhinitis, is a metabolic derivative of another second-generation drug, loratadine[46]. For this reason, desloratadine has been called a new-generation antihistamine; however, except for displaying a higher affinity and slower disassociation rate with the target receptor H₁R[47], desloratadine does not appear to confer improved efficacy or additional clinical benefit compared to loratadine. Nevertheless, it is still of great significance to uncover the molecular mechanism leading to the difference in affinity to provide clues for the design of next-generation antihistamines.

The solved structure of desloratadine-H₁R shows that desloratadine is buried in the main orthosteric pocket, and its tricyclic group is enclosed by a series of hydrophobic residues such as Y108[3.33], Y431[6.51] and F432[6.52] in a manner similar to the other two ligands (Fig. 2f). In addition to the hydrophobic interactions, Y431[6.51] also forms a hydrogen bond with the N atom in the pyridine moiety, as in the mepyramine-H₁R complex (Fig. 2f). Mutation of Y431[6.51] and F432[6.52], two residues in which mutation does not significantly affect the basal signaling ability of H₁R[34,48], dramatically impairs the ability of mepyramine and desloratadine to inhibit H₁R signaling (Supplementary Fig. 8 and Supplementary Table 2), demonstrating the indispensability of main pocket residues in the coordination of ligands. Finally, the

piperidine ring also forms an electrostatic interaction with D107[3.32] (Fig. 2f). Interestingly, the only difference between desloratadine and loratadine is that the ethoxycarbonyl group attached to the piperidine ring in loratadine is replaced with hydrogen in desloratadine. This substitution causes desloratadine to have approximately two orders of magnitude higher affinity for H₁R than loratadine. To uncover the underlying mechanism, we again removed the desloratadine molecule and docked loratadine onto this structure (Supplementary Table 3). As anticipated, the tricyclic group of loratadine is located in the main pocket, similar to desloratadine, in the docked model. The ethoxycarbonyl group extends into the secondary binding pocket (Fig. 2g). However, unlike the methoxyphenyl moiety in astemizole, the ethoxycarbonyl group is much smaller in size and cannot form effective hydrophobic contacts with the surrounding residues Y87[2.64], W103[3.28], and M451[7.36]. Instead, the carbonyl tail even faces two hydrophilic atoms from H₁R residues N84[2.61] and Y458[7.43] (Fig. 2g). Obviously, when the tricyclic group of loratadine is positioned in the main ligand binding pocket, very reasonable accommodation of the ethoxycarbonyl group in the secondary binding pocket will not be achieved. Consistently, the measured potency of loratadine in inhibiting H₁R was much lower than that of desloratadine (Fig. 2h and Supplementary Table 2). Structure-guided introduction of hydrophilic derivatives into loratadine at the corresponding position of the ethoxycarbonyl group may result in antihistamines with better clinical performance.

### Comparison of the ligand pocket in different states

With the availability of H₁R structures bound to different ligands, we next performed a detailed comparison of the ligand pocket shaped by different agents. Superposition of the H₁R structures bound to second-generation antihistamines from this study reveals that the residues lining the entire ligand pocket adopt roughly similar configurations (Fig. 3a), which suggests that H₁R may employ a general recognition mechanism to engage antihistamines, especially when the antihistamines share a similar chemotype. However, when comparing the astemizole-bound ligand pocket to that bound with the first-generation antihistamine doxepin, remarkable conformational differences were observed in the region surrounding the secondary pocket (Fig. 3b). In the doxepin-bound structure, H₁R ECL2 occupies the vestibule above the secondary pocket, but in the astemizole-bound structure, ECL2 adopts a lifted conformation, releasing more space for accommodation of the methoxyphenyl moiety of astemizole (Fig. 3b). Meanwhile, the side chain of Y87[2.64] also rotates to participate in the construction of the secondary pocket (Fig. 3b). Further comparison of

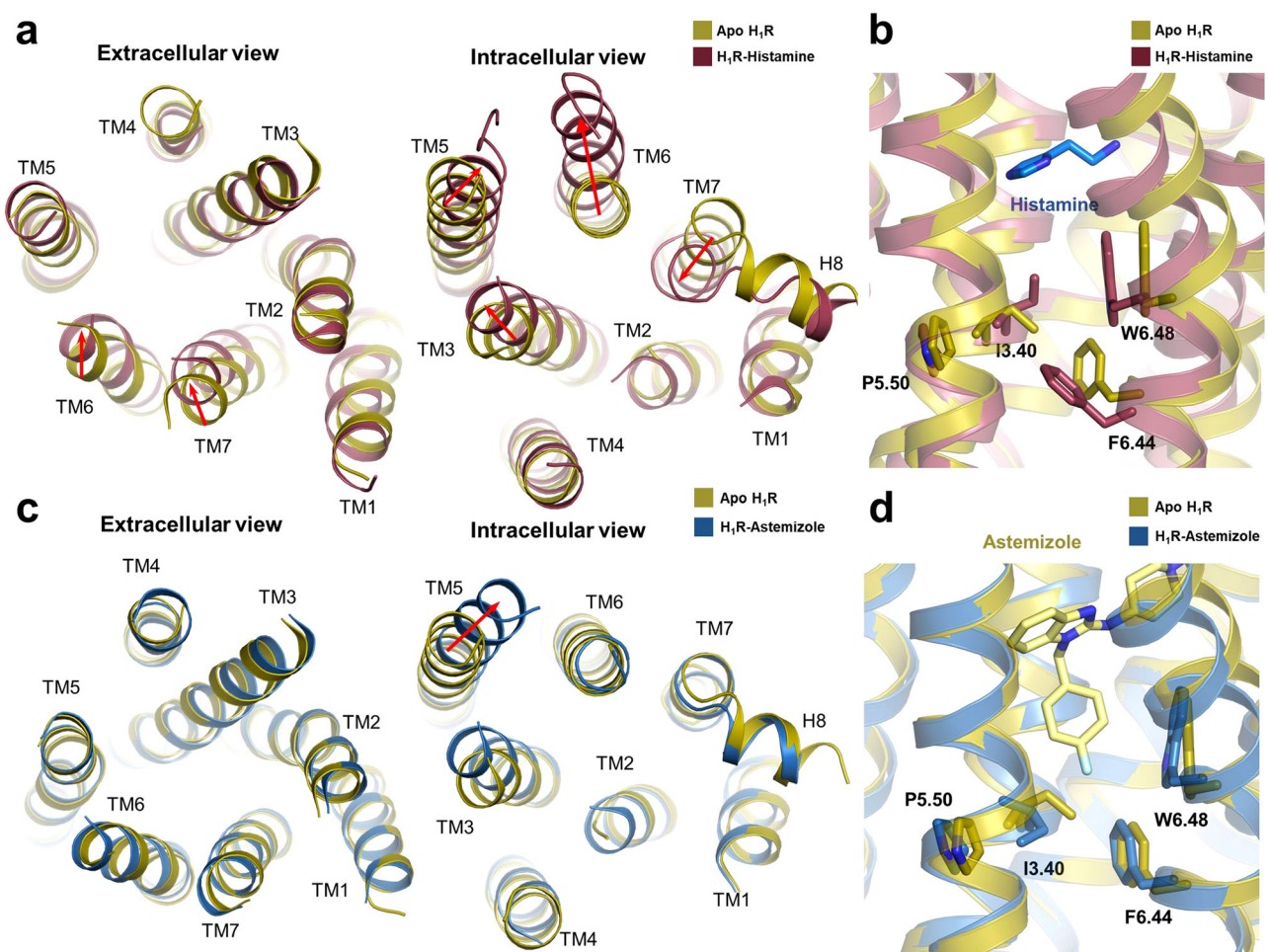

**Fig. 4 | Conformational changes during H₁R activation. a** Structural comparisons of H₁R bound to histamine (PDB ID: 7DFL) and in apo H₁R. Significant conformational changes are indicated with red arrows. **b** Comparison of the $P^{5.50}$-$I^{3.40}$-$F^{6.44}$ triadmotif and $W^{6.28}$ in the H₁R apo form (olive) and histamine-bound active form (raspberry). Histamine is shown as marine sticks. **c** Structural comparisons of H₁R bound to astemziole (sky blue) and in apo form (olive). Significant conformational changes are indicated with red arrows. **d** Comparison of the $P^{5.50}$-$I^{3.40}$-$F^{6.44}$ triadmotif and $W^{6.28}$ in the H₁R apo form (olive) and the astemizole-bound H₁R inactive form (sky blue). Astemizole color as pale yellow.

the astemizole-bound H₁R with that bound to the endogenous agonist histamine revealed a more extensive conformational change in the ligand pocket (Fig. 3c). Due to the smaller size of the agonist histamine, the active pocket becomes more constricted to make full contact with the ligand. For example, residues $Y431^{6.51}$, $F432^{6.52}$, $H450^{7.35}$ and $Y458^{7.43}$ in the active structure all move closer toward the receptor core (Fig. 3c). Again, ECL2 sits on top of the secondary pocket, although it is not inserted as deep into the pocket as that in the doxepin-bound structure (Fig. 3c). Together, these results demonstrate that there is structural plasticity in the ligand-binding pocket of H₁R, and it is worth making full use of these different states of the pocket for the development of next-generation drugs.

## Mechanism of H₁R activation

An important property of H₁R is its high basal signaling capacity, a distinctive feature observed in many GPCRs. Due to the lack of efficient methods to solve the structure of GPCRs in the apo state, whether these GPCRs maintain an activity-like structure or what conformation they have in the apo state remains unknown. Enabled by our recently developed method, we successfully solved the structure of H₁R in the apo state (Fig. 1d). The determined structure shows that it adopts a typical inactive conformation in the apo state (Fig. 1d). Therefore, the high basal signal of H₁R may not be attributed to the preexisting stable open conformation of TM6. Interaction with a G protein appears to still

be required to stabilize H₁R in a fully active state. Although a 'squash to activate and expand to deactivate' model has been proposed for the activation of H₁R through structural comparison of inverse agonist-bound and agonist-bound H₁R[48], the existence of an additional apo structure provides a more comprehensive understanding of H₁R activation and regulation (Fig. 4). In the apo state, TM6 and TM7 appear to adopt an intermediate conformation, and this conformation is induced to move in opposite directions by inverse agonists and agonists (Supplementary Fig. 9). Specifically, histamine binding induces a further contraction of TM6 and TM7 on the extracellular side (Fig. 4a). The resulting closed packing on the extracellular region drives the downward movement of key activation motifs, the toggle residue $W428^{6.48}$ and the $P^{5.50}$-$I^{3.40}$-$F^{6.44}$ triadmotif (Fig. 4b). These conformational rearrangements then propagate to the intracellular side and result in the outward movement of TM6 (Fig. 4a). In contrast, engagement of an inverse agonist at the orthosteric site seems to insulate the contact between TM3 and TM6 and results in a wider opening of the extracellular core of the agonist-stabilized structures (Fig. 4c). Therefore, the ready accessibility between TM3 and TM6 in the apo state of H₁R is probably one of the reasons for its basal signaling ability. Next, we probed the mechanism of inverse modulation of histamines. Taking the H₁R structure bound to astemizole as an example, the fluorophenyl group of astemizole inserted deeply into the vicinity of $W428^{6.48}$ and completely prevented its downward

movement (Fig. 4d). As a result, the $W428^{6.48}$ and $P^{5.50}$-$I^{3.40}$-$F^{6.44}$ triad-motif are constantly fixed in inactive positions, leading to the closed packing of TM6 against the intracellular core and ultimately preventing basal activation of $H_1R$ (Fig. 4d). Therefore, all these compounds are classified as inverse agonists. In summary, agonist and inverse agonist binding to $H_1R$ induces an opposite conformational change in the extracellular region, which ultimately causes different signaling outputs on the intracellular side.

## Discussion

Antihistamines that block $H_1R$ activity are commonly used to prevent and treat symptoms of allergic rhinitis, allergic conjunctivitis, and urticaria[12]. Although second-generation antihistamines have offered many advantages over first-generation antihistamines, such as minimal blood–brain barrier penetration, efforts are still being made to develop newer generations of antihistamines with higher efficacy and safety[14]. In recent years, structure-guided drug screening and design has become an effective strategy in the field of GPCRs[49–52]; however, access to different structural states of target receptors is often a prerequisite. In this regard, a detailed analysis of the recognition mechanism between $H_1R$ and known antihistamines will be of great significance and can provide valuable clues for the development of a new generation of drugs. Therefore, in this study, we performed structural studies on $H_1R$ in complex with three representative antihistamines, mepyramine, astemizole and desloratadine. In these structures, a secondary pocket composed of ECL2, TM2, TM3 and TM7 was identified in addition to the conventional orthosteric site (Fig. 2b). Unlike the hydrophobic feature of the orthosteric site, this secondary pocket is decorated with several polar residues. Therefore, the introduction of optimal derivative groups into existing antihistamines can be explored in the future.

All four histamine receptors display constitutive activity[11]; however, the ligands for each receptor subtype have distinct pharmacological profiles[53]. Of these, all validated $H_1R$ blockers act as inverse agonists, in contrast to the existence of neutral antagonists for the other three histamine receptors[54–56]. Examination of all reported $H_1R$ blockers shows that almost all of them contain a phenyl group at the hydrophobic core (Supplementary Fig. 10), and according to the structural information in this study, this phenyl group is inserted into the deep cavity and will block the activation switch of the toggle switch residue $W428^{6.48}$. As a result, all these $H_1R$ blockers most likely function as inverse agonists as well. Comparison of the ligand pocket of $H_1R$ with those of the other three histamine receptors reveals clear differences in the main pocket, which should result in the specificity profile of various antihistamines. For example, residues 3.33, 5.46 and 6.52 make close contact with the three antihistamines used in this study; however, they exhibit sequence divergence in the other three histamine receptors (Supplementary Fig. 11). This difference may account for the exclusive selectivity for $H_1R$ of mepyramine, astemizole, and desloratadine. Detailed comparison of ligand pockets, including those from aminergic receptors, further reveals that the secondary ligand pockets are even less conserved (Supplementary Fig. 12). Therefore, taking full advantage of the features of secondary pockets may lead to the development of new antihistamine drugs that are more selective and effective. Consistent with the different configurations of the ligand pocket, the common ligands for each receptor subtype also have accordingly different chemical structures (Supplementary Fig. 13). For example, the ligands for $H_2R$ and $H_3R$ have a more linear structure[57]. Furthermore, their binding sites to $H_2R$ and $H_3R$ are not as deep as those involved in $H_1R$ (Supplementary Fig. 14); therefore, these ligands cannot directly impose conformational constraints on the toggle switch residue. As a result, they cannot employ the $H_1R$-ligand-like mechanism to modulate the target receptors, and thus, different modulation mechanisms, such as neutral antagonism, emerge[56,58]. Taken together, the results here provide insights into the molecular regulation of $H_1R$ inverse agonists and a framework for optimizing a new generation of antihistamines.

## Methods

### Expression and purification of Fab and glue molecules

The coding sequences of the Fab or glue molecule were cloned and inserted into the pET-22b (+) vector with an N-terminal pelB signal peptide and a C-terminal 6xHis tag. Plasmids were transformed into *E. coli* BL21 (DE3) cells. Cells were grown to $OD_{600} = 0.8$ at 37 °C in LB medium containing 1‰ ampicillin, and a final concentration of 0.5 mM isopropyl-β-D-thiogalactopyranoside (IPTG) was added to the medium and then cultured for 18 ~ 24 hours at 16 °C. Cells were harvested and disrupted by sonication. Both Fab and glue molecules were purified by Ni-NTA chromatography. The unwanted proteins were removed with wash buffer (20 mM HEPES pH 7.5, 200 mM NaCl, 20 mM imidazole), and then the target protein was eluted with wash buffer supplemented with 300 mM imidazole. The elution was concentrated using a 10-kDa molecular weight cutoff concentrator (Millipore). The concentrated proteins were aliquoted, flash frozen in liquid nitrogen and stored at −80 °C before use.

### Construct design of $H_1R$-mBRIL

The codon-optimized human $H_1R$ gene was cloned and inserted into the pFastbac A vector (Thermo Fisher Cat# 10360014) with a hemagglutinin (HA) signal peptide followed by a FLAG tag at the N-terminus and a 10× His tag at the C-terminus. ICL3 residues 217-408 and C-terminal residues 484-487 of $H_1R$ were deleted and replaced by mBRIL and K3-ALFA tags, respectively.

### Complex formation and purification

Recombinant baculovirus for insect cell expression was made using the Bac-to-Bac baculovirus expression system (Thermo Fisher Cat# 10360014). *Spodoptera frugiperda Sf9* cells (Invitrogen Cat# A35243) were grown in SIM SF Medium (Sino Biological Inc.) at 27 °C and were infected with recombinant baculovirus containing the $H_1R$-mBRIL gene at a density of $4 \times 10^6$ cells per mL. After 48 hours of infection, the cells were spun down, and cell pellets were stored at −80 °C until use.

Thawed cell pellets were resuspended in lysis buffer composed of 10 mM HEPES pH 7.5, 1 mM PMSF, and 0.5 mM EDTA. Cell membranes were then spun down and solubilized with a buffer of 20 mM HEPES pH 7.5, 500 mM NaCl, 5 μM mepyramine maleate (inverse agonist, TOPSCIENCE Cat# T1232), 1% (w/v) n-dodecyl-B-D-maltoside (DDM, Anatrace Cat# D310), and 0.1% (w/v) cholesteryl hemisuccinate (CHS, Sigma Cat# C6512) at 4 °C for 2 hours. The solubilized receptor was isolated by centrifugation and incubated with Ni-NTA chromatography at 4 °C for 2 hours. The resin was collected in a column, washed with a buffer composed of 20 mM HEPES pH 7.5, 150 mM NaCl, 5 μM mepyramine maleate, 0.03% (w/v) DDM, and 0.02% (w/v) CHS, and eluted by wash buffer supplemented with 250 mM imidazole. Then, 2 mM $CaCl_2$, excess purified Fab and glue molecules were added to the elution and incubated with the anti-FLAG M1 affinity resin (M1 resin, Sigma–Aldrich Cat# A4596) at 4 °C for 1 hour. M1 resin was then collected and washed with a buffer containing 20 mM HEPES pH 7.5, 150 mM NaCl, 5 μM mepyramine maleate, 0.03% DDM, 0.01% CHS, and 2 mM $CaCl_2$. The complex was then gradually exchanged into a buffer containing 20 mM HEPES pH 7.5, 150 mM NaCl, 5 μM mepyramine maleate, 0.1% (w/v) LMNG, 0.01% (w/v) CHS, and 2 mM $CaCl_2$ and then eluted with a buffer containing 20 mM HEPES pH 7.5, 150 mM NaCl, 5 μM mepyramine maleate, 0.00075% (w/v) LMNG, 0.00025% (w/v) glycol-diosgenin (GDN, Anatrace Cat#GDN101), 0.0001% CHS, 5 mM EDTA and 200 μg/ml synthesized Flag peptide. The complex was further purified by size-exclusion chromatography using a Superdex 200 Increase 10/300 column (GE Healthcare) preequilibrated with buffer containing 20 mM HEPES pH 7.5, 150 mM NaCl, 5 μM mepyramine maleate, 0.00075% (w/v) LMNG, 0.00025% (w/v) GDN, and 0.0001%

(w/v) CHS. The monodisperse peak fractions were collected and concentrated to ~5 mg/ml for cryo-EM analysis. For the $H_1R$-mBRIL complex bound to astemizole and desloratadine, 10 μM astemizole (inverse agonist, TOPSCIENCE Cat# T1278) or 10 μM desloratadine (inverse agonist, TOPSCIENCE Cat# T2520) was added at each step during purification.

## Cryo-EM sample preparation and data acquisition

To prepare the cryo-EM grids of the $H_1R$-mBril complexes, 3.5 μL of sample was applied onto a glow-charged amorphous alloy film grid (CryoMatrix nickel titanium alloy film, R1.2/1.3, Zhenjiang Lehua Electronic Technology Co., Ltd.)[59]. The grids were vitrified in liquid ethane using a Vitrobot Mark IV (Thermo Fisher Scientific) instrument with a blot force of 4, blot time of 4 s, humidity of 100%, and temperature of 8 °C. Grids were first screened on an FEI 200 kV Arctica transmission electron microscope (TEM), and grids with evenly distributed thin ice and promising were transferred to an FEI 300 kV Titan Krios TEM (Thermo Fisher Scientific FEI, the Center for Integrative Imaging, Hefei National Laboratory for Physical Sciences at the Microscale, University of Science and Technology of China) equipped with a Gatan Quantum energy filter and a spherical corrector for data collection. Images were taken by a Gatan K3 direct electron detector at a magnitude of 81,000, superresolution counting model at a pixel size of 0.535 Å. Each image was dose-fractionated in 32 frames using a total exposure time of 4.1 s at a dose rate of 14.174 e/pixel/second. All image stacks were collected by the EPU program of FEI, and the nominal defocus value varied from −1.2 to 2.0 μm.

## Cryo-EM data processing

For the apo $H_1R$-mBRIL complex, mepyramine-$H_1R$-mBRIL complex, astemizole-$H_1R$-mBRIL complex and desloratadine-$H_1R$-mBRIL complex, 4536, 3644, 3572 and 4066 movies were collected, respectively, and then binned 2-fold in cryoSPARC v.3.2.0[60] using patch motion correction, yielding a pixel size of 1.07 Å. Contrast transfer function (CTF) parameters for each micrograph were estimated by patch CTF estimation in cryoSPARC. A total of 4739217, 4085241, 4034639, and 4562548 particle projections were produced by autopicking the apo $H_1R$-mBRIL complex, mepyramine-$H_1R$-mBRIL complex, astemizole-$H_1R$-mBRIL complex and desloratadine-$H_1R$-mBRIL, respectively. Then, all particles were used to perform several cycles of 2D classification and 3D classification to discard false-positive particles. The final datasets of 327890, 387910, 357476 and 425634 particle projections from the best class were further applied for final homogenous refinement, particle subtraction and local refinement in cryoSPARC, and density maps were obtained with nominal resolutions of 3.5 Å, 3.2 Å, 3.0 Å and 3.4 Å (determined by FSC using the 0.143 criterion) for the apo $H_1R$-mBRIL complex, mepyramine-$H_1R$-mBRIL complex, astemizole-$H_1R$-mBRIL complex and desloratadine-$H_1R$-mBRIL complex, respectively.

## Model building and refinement

The crystal structures of human doxepin-bound $H_1R$ (PDB ID: 3RZE) were used as initial models for model rebuilding and refinement against the electron microscopy map. The structure of E3/K3 in the complexes was from the structure of the E3/K3 coiled coil (PDB ID: 1U0I). The ALFA tag and ALFA-Nb were obtained from the crystal structure of ALFA-Nb bound to the ALFA-tag peptide (PDB ID: 6I2G). The structure of Fab-Nb in the complexes was from the crystal structure of pinatuzumab Fab with an anti-Kappa VHH domain (PDB ID: 6AND). All the models were docked into the EM density map using Chimera[61], followed by iterative manual adjustment and rebuilding in COOT[62] and phenix real_space refine in Phenix[63]. The final model statistics were validated using MolProbity[64]. Model refinement statistics are summarized in Supplementary Table 1. The molecular graphic figures were prepared with UCSF ChimeraX[65] and PyMOL.

## Molecular docking

Molecular docking was performed using Auto Dock Tools (ADT) (version 1.5.7)[66] and Auto Dock Vina (version 1_1_2 docking programs)[67] to understand the drug molecule interaction with the protein, the potential binding mode, and energy. The structure of $H_1R$ reported here was used as the receptor, and the structures of antagonists downloaded from the PubChem database were used as ligands. The receptor and ligands were prepared by AutoDockTools to produce the corresponding low-energy three-dimensional conformation and the correct ionization state (pH 7.0). A 3D docking grid centered in the $H_1R$ structure was generated, and residues around the pocket were treated as flexible. Then, the processed inverse agonists were docked into the binding pocket of $H_1R$, outputting the top 10 conformations for each ligand. The most reliable binding poses were selected according to the interaction energy and visual inspection. All results were analyzed and visualized using PyMOL (http://www.pymol.org).

## Inositol phosphate accumulation assay

IP1 production was detected by using the IP-One $G_q$ HTRF kit (Cisbio). HEK293 cells were seeded into 24-well culture plates (Corning) at a density of 0.1 million per well and incubated overnight at 37 °C with 5% $CO_2$. Plasmids expressing wild-type $H_1R$ or its mutants were transiently transfected using Lipofectamine™ 3000 regent (Invitrogen) when the cells reached 75% confluence. Twenty-four hours after transfection, the culture media was removed, and transfected cells were harvested and washed with Dulbecco's phosphate buffered saline (DPBS) buffer (Gibco) twice and then resuspended in Hank's balanced salt solution (HBSS) buffer (Beyotime) at a density of $1.0 \times 10^6$ cells per milliliter. The 7 μL cell resuspension was seeded into a 384-well plate (Perkin Elmer) and incubated with 7 μL agonist or inverse agonist with various concentration gradients for 1 hour at 37 °C. Afterward, 3 μL IP1 d2 reagent and IP1 Tb cryptate antibody were added to the 384-well plate and incubated for another 1 h at room temperature. Then, an EnVision multimode plate reader (Perkin Elmer) was employed to measure the HTRF ratio at 620/665 nm. The accumulation of IP1 was calculated according to a standard dose–response curve in GraphPad Prism 8 (GraphPad Software). Data are represented as the mean ± SEM, n = 3 independent samples.

## Reporting summary

Further information on research design is available in the Nature Portfolio Reporting Summary linked to this article.

# Data availability

The structural data generated in this study have been deposited in the Electron Microscopy Database (EMDB) and the protein data bank (PDB) with the following accession codes: EMD-38078 and 8X63 [https://doi.org/10.2210/pdb8X63/pdb] for the mepyramine-$H_1R$; EMD-38075 and 8X5Y for the astemizole-$H_1R$; EMD-38079 and 8X64 for the desloratadine-$H_1R$; EMD-38074 and 8X5X for the apo $H_1R$. The PDB datasets used for analysis in this study include 3RZE, 7DFL, 7UL3, 7F61, 1U0I, 6I2G and 6AND. All the other data generated in this study are provided in the Supplementary information and source data files. Source data are provided with this paper.

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

## Acknowledgements

The cryo-EM data were collected at the Center of Cryo-Electron Microscopy, University of Science and Technology of China with the help of Dr. Yongxiang Gao. This work was supported by the National Key Research and Development Program of China Grants (2022YFF1203100, 2018YFA0902700 to Y.T. and 2022YFC2303700, 2022YFA1302700 to K.Z.), Anhui Provincial Natural Science Foundation 2308085QC77 (to Q.G.), the Strategic Priority Research Program of the Chinese Academy of Sciences (XDB0490000 to K.Z.) and the Center for Advanced Interdisciplinary Science and Biomedicine of IHM (QYZD20220006 to Y.T. and QYPY20220019 to K.Z.).

## Author contributions

D.W. and Q.G. designed, expressed, purified, and prepared the cryo-EM samples. D.W. and B.H. collected the cryo-EM data and helped with data processing. M.L. and K.Z. processed the cryo-EM data. D.W., Z.W. and Y.D. conducted the biochemical experiment. Y.T. conceived and designed the overall project, supervised the research, and wrote the manuscript with input from all co-authors.

## Competing interests

The authors declare no competing interests.
