## [Peer Review File · Nature Communications]

Molecular mechanism of antihistamines recognition and regulation of the histamine H1 receptorREVIEWER COMMENTS

Reviewer #1 (Remarks to the Author):

This is an interesting and important paper, though I have a few concerns to improve the manuscript and enhance its impact.

Major points:

- 1) For all three ligands in the H1 receptor structures, the cell-based functional assays should be performed including the mutagenesis assays of residues involving the ligand-binding. Also, the comparison of the affinity of the ligands should be calculated in dose-dependent biochemical assays.
- 2) For the three structural models for apo, mepyramine, and desloratadine-bound H1R, the clash score and MolProbity score in supplementary table 1 are too high to be published. It should be built and refined again with reliable quality to be published. All structural analyses should be performed again based on the re-built models.

Minor points:

- 1) Supplementary Fig. 2-5: The FSC curve for the locally refined 3D reconstruction on the TM should be presented for validation. Local resolution analysis for all the 3D reconstructions should be performed and presented.
- 2) Line 131: "...H1R may have a dynamic structure, reminiscent of its high basal signaling capacity.", it would be helpful to present more specifically using 3D variability analysis in CryoSPARC.
- 3) Supplementary Fig.6a: The label F4246.48 should be corrected as F4246.44.
- 4) Line 222: "Secondary binding pocket" has been ambiguously described. It needs to be specifically indicated using a circle in Fig.2b as compared with the main ligand binding pocket.
- 5) Line 237-246: The interactions in the Mizolastine-docked model should be experimentally validated based on biochemical assays.
- 6) Line 273-280: The interactions in the Loratadine-docked model should be experimentally validated based on biochemical assays.
- 7) Line 283: "...loratadine does not possess a high affinity", how to define a high affinity? If it means relatively higher affinity compared to Desloratadine, the affinity of Desloratadine and Loratadine against the H1 receptor should be compared based on the biochemical assays.
- 8) Line 481-500: The references of the software for cryo-EM data processing should be added.
- 9) Supplementary Table 1: In the Ramachandran plot for the Mepyramine-H1R-mBRIL model, the Favored and Allowed percentages might be changed.

Reviewer #2 (Remarks to the Author):

In this study, the authors made great structural efforts in understanding the molecular recognition and regulation of histamine H1 receptor (H1R) by divergent antihistamines. Using a mBril fusion and gluing strategy, the authors determined the cryo-EM structures of H1R bound to three inverse agonists, mepyramine, astemizole and desloratadine, as well as H1R in apo state. Structural analysis uncovered another extended ligand binding pocket in H1R when occupied by astemizole, and revealed the structural determinants about how these three agonists exert inverse agonistic activity. Overall, this is a nice work which provides new insights into the ligand binding and inverse agonism of H1R by antihistamines. However, the quality of this paper is restricted by the lack of pharmacological profiling

and the limitation of the methodology regarding to the structure determination. My comments:

Major concerns:

1. The authors showed the recognition details of mepyramine, astemizole and desloratadine in H1R. In the main orthosteric pocket of H1R, although these ligands share a similar set of interaction residues, differences in the residue composition and conformation could be observed, which may account for their distinct ligand potency. The authors also declared that the proper interaction of the ligand with extended pocket of H1R occupied by methoxyphenyl moiety of astemizole is critical for the ligand affinity and specificity. However, this point is also not supported by any pharmacological data. Would mutation of the polar residues K179, Y87 or N84 in the extended pocket into hydrophobic residues increase the potency of astemizole to H1R? Additional mutational assays using either IP1 accumulation or radioligand binding would be encouraged to verify the ligand poses and further clarify the detailed structural determinants in the main orthosteric pocket and the extended pocket for ligand potency.

2. H1R has high basal activity. It is interesting to see that the apo H1R structure reported in this study is in its inactive conformation, as the TM6 inwardly moved. It well known in GPCR area that the outward kink of TM6 is critical feature of GPCR activation, which is necessary for the opening of the receptor intracellular cavity for G protein coupling. As the inactive apo H1R conformation is not favorable for G protein coupling, one would think this inactive apo conformation maybe caused by the tight tethering of the mBril fusion and the Helix8 of H1R, rather than the real apo state of the receptor. Thus, the authors should tune down the discussion on the mechanism of basal activity of H1R and add more discussion on the limitation of the methodology in determination of apo GPCR structure.

3. All the three ligands, mepyramine, astemizole and desloratadine, behave as selective H1R inverse agonists. More structural comparison of these inverse agonists bound H1R structures with other histamine receptor subtypes (H2R, H3R and H4R) would further elucidate the ligand selectivity determinants of histamine receptor family.

Minor issues:

1. Line 161: change 'orthosteric pocket mainly constituted by TM3...' into 'orthosteric pocket mainly constituted by residues from TM3...'
2. Line 181: 'As anticipated' should be 'as anticipated'
3. Line 221: change 'defined by ECL2...' into 'defined by residues from ECL2...'
4. Line 230: 'promises to' is not properly used in this sentence, please correct it.
5. Some H1R ligands such as oxatomide were reported to be neutral antagonists of H1R, are there any differences in the ligand binding between H1R neutral antagonist and inverse agonist? Perhaps the authors could do some molecular docking work to gain more insights into the ligand binding property of H1R.

Reviewer #3 (Remarks to the Author):

In this study, Wang, et al. present four cryo-EM structures of H1R in apo and antihistamine-bound states. These structures show the H1R binding details of three inverse agonists, including first-generation antihistamine mepyramine and second-generation ones astemizole and desloratadine. They also identify a secondary ligand-binding site that may support the introduction of new derivative groups to generate newer antihistamines. The activation mechanism of H1R is also been discussed. This work is of importance in understanding antihistamine recognition by H1R and guiding new-generation antihistamines targeting H1R. However, there are still issues to be addressed.

Major concerns:

1. The clash scores for the mepyramine- and desloratadine-bound H1R structures exceed 36 and 47,

respectively. It is strongly recommended to further optimize these models to obtain more reliable and convincing models. Additionally, there seems to be an inversion in the data for favored (%) and allowed (%), which should be rectified.

2. -Lines 100-116, since reference 28 has not been published, the audiences may be confused by the gluing strategy and further the design of "4-9" and "6-13" glues. To ensure clarity, it is essential to provide the necessary information to elucidate the gluing strategy and the design of these specific glues.

3. -Lines 130 and 131, the statement that "without inverse agonist stabilization, H1R may have a dynamic structure, reminiscent of its high basal signaling capacity" is incorrect. In the apo state, most GPCRs exist in an equilibrium between different conformational states. This equilibrium is why only a few GPCR structures have been solved without ligands or downstream signaling proteins. Therefore, it is not convincing to correlate the lower density maps and dynamic conformation of apo H1R with its "high basal signaling capacity."

4. -Lines 228-246, the author's assertion regarding 'modifying the methoxyphenyl moiety with alternative derivative groups, for example, introducing hydrogen bond donors/acceptors, may further improve the affinity and specificity of astemizole' is not well supported by the subsequent description. It is important to address whether there are any differences in ligand affinity between mizolastine and astemizole. Additionally, it would be valuable to discuss whether the sequence differences at the secondary pocket across the four histamine subtypes support the high selectivity of mizolastine over H1R. Further discussions on these issues would provide stronger support for the author's hypothesis.

5. Given that the identification of the second binding pocket and its potential contribution to subtype selectivity constitutes a crucial aspect of this study, it is imperative to conduct mutagenesis assays and other functional analyses, rather than relying on molecular docking, to provide robust evidence supporting the author's hypothesis.

6. -Fig. 2c,e, it appears that mizolastine and loratadine, in the docked models, penetrate deeper into the binding site compared to astemizole and desloratadine in the experimental models, respectively. For instance, D1073.32 establishes an electrostatic interaction with the N atom in the piperidine ring of desloratadine, whereas it interacts with the O atom in the carboxylate group of loratadine positioned above the piperidine ring. Has the author attempted to merge the experimental and docked models to compare the experimental and docked models? It would be intriguing to explore whether they exhibit different binding depths despite sharing the same tricyclic chemical group.

7. -Lines 326-329, the author should be very careful to conclude 'it adopts a typical inactive conformation in the apo state'. The fusion protein attaching to ICL3 may modify the equilibria between different conformational states (Eddy, M. T., et al. *Structure* 24, 2190–2197, 2016), especially the two rigid linkers between TM5, TM6, and N-, C-term of BRIL, respectively. That is, the glue strategy may strongly restrict and force H1R to the inactive state to leading false judgment.

8. -Lines 320-350, the reviewer noticed that a 'squash to activate and expand to deactivate' model has been proposed for H1R previously (Xia, et al., *Nat Commun.* 2021, 12(1):2086). According to this model, the salt-bridge interaction between TM3 and TM6 is suggested to mediate the basal activity of H1R, while the bulky tricyclic groups of inverse agonists induce an inactive conformation by pushing phenylalanine in TM6. The question arises whether the author's findings align with these previous conclusions or provide new insights into H1R activation. It is important to include a discussion addressing this point and discussing the consistency or novelty of the author's findings concerning the previously proposed model.

9. -Lines 375-377, reference 52 is incorrect. Please ensure the correct reference is cited. Additionally, based on information from the IUPHAR database (<https://www.guidetopharmacology.org/>), it is stated that blockers for all four histamine receptor subtypes function as both inverse agonists and neutral antagonists. It is recommended to double-check this information for accuracy.

10. -Lines 382-390, to the reviewer's knowledge, there is no evidence supporting the claim that the direct interaction between blockers and the toggle switch W6.48 can distinguish inverse agonists from neutral antagonists. Please cite corresponding references to support the author's statement.

Minor concerns:

1. -Fig.1, the EM density of several helices of H1R is shown discontinuous. Please adjust the EM

threshold. In addition, the names of ligands are suggested to be labeled and the sticks of ligands should be colored by heteroatom.

2. -Fig.2a, W4326.52 should be F4326.52.

3. -Fig.2a,e, W1083.33 should be Y1083.33.

4. -Fig. S8, H3R-PF-03654746 (PDB ID: 7D6L) is incorrect, which should be 7F61.

5. -Line 427, sf9 insect cells at a density of 4×10^3 per ml are unreasonable, please double-check.

6. -Line 463, 'complexex' should be 'complexes'.

Reviewer #1 (Remarks to the Author):

This is an interesting and important paper, though I have a few concerns to improve the manuscript and enhance its impact.

Thanks to Reviewer #1. We appreciate very much that Reviewer #1 thinks our work provides valuable information for the field.

Major points:

1) For all three ligands in the H₁R receptor structures, the cell-based functional assays should be performed including the mutagenesis assays of residues involving the ligand-binding. Also, the comparison of the affinity of the ligands should be calculated in dose-dependent biochemical assays.

We have conducted H₁R signaling assays including several mutants focused on the secondary pocket. The results are provided in revised Fig. 2c, 2e and 2h. These functional data support the structural observations; therefore, the conclusion of this study is further confirmed.

2) For the three structural models for apo, mepyramine, and desloratadine-bound H₁R, the clash score and MolProbity score in supplementary table 1 are too high to be published. It should be built and refined again with reliable quality to be published. All structural analyses should be performed again based on the re-built models.

Thanks to Reviewer #1. We have refined the mentioned structures and performed structural analysis.

Minor points:

1) Supplementary Fig. 2-5: The FSC curve for the locally refined 3D reconstruction on the TM should be presented for validation. Local resolution analysis for all the 3D reconstructions should be performed and presented.

We have provided the suggested information in the revised Supplementary Fig. 2-5. Specifically, the FSC curve for the 3D reconstruction and the resolution maps for the final 3D reconstruction are provided in the revised Supplementary Fig. 2e-5e and Supplementary Fig. 2d-5d, respectively.

2) Line 131: "...H₁R may have a dynamic structure, reminiscent of its high basal signaling capacity.", it would be helpful to present more specifically using 3D variability analysis in CryoSPARC.

We conducted a 3D variability analysis. As an example, two density maps reconstituted from two classes are shown in Supplementary Fig. 6 (also below), which demonstrates the high flexibilities in the TM region in apo H₁R. Furthermore, to be more accurate, we have changed the description in the revised manuscript.

3) Supplementary Fig.6a: The label F4246.48 should be corrected as F4246.44.

Thank you (now Supplementary Fig. 7a in the revised version).

4) Line 222: “Secondary binding pocket” has been ambiguously described. It needs to be specifically indicated using a circle in Fig.2b as compared with the main ligand binding pocket.

We have modified the corresponding figure as suggested.

5) Line 237-246: The interactions in the Mizolastine-docked model should be experimentally validated based on biochemical assays.

We have measured the ability of mizolastine to inhibit the signaling of wild-type H₁R and mutant H₁R. Based on the docking model, H450^{7.35} from the secondary pocket forms a hydrogen bond with the dihydropyrimidine group of mizolastine (Fig. 2d). Consistently, mutation of H450^{7.35} indeed crippled the ability of mizolastine to inhibit H₁R signaling (Fig. 2e). In contrast, H450^{7.35} does not form any hydrogen bond with astemizole; furthermore, it is even incompatible with the hydrophobic feature of astemizole, so the H450^{7.35} mutation increases the inhibition efficacy of astemizole (an opposite behavior with respect to mizolastine) (Fig. 2c). The direct comparison is shown below.

6) Line 273-280: The interactions in the Loratadine-docked model should be experimentally validated based on biochemical assays.

We have measured the inhibitory ability of desloratadine and loratadine on H₁R. As loratadine contains an additional ethoxycarbonyl group that is environmentally incompatible (as described in the manuscript), its inhibitory potency is much lower than desloratadine (shown in the revised Fig. 2h and also below).

7) Line 283: "...loratadine does not possess a high affinity", how to define a high affinity? If it means relatively higher affinity compared to Desloratadine, the affinity of Desloratadine and Loratadine against the H₁ receptor should be compared based on the biochemical assays.

We have modified the description in the revised manuscript. Functional assays also confirmed that loratadine is less potent than desloratadine, as shown in Fig. 2h.

8) Line 481-500: The references of the software for cryo-EM data processing should be added.

We have added the references.

9) Supplementary Table 1: In the Ramachandran plot for the Mepyramine-H₁R-mBRIL model, the Favored and Allowed percentages might be changed.

Thanks a lot.

Reviewer #2 (Remarks to the Author):

In this study, the authors made great structural efforts in understanding the molecular recognition and regulation of histamine H1 receptor (H1R) by divergent antihistamines. Using a mBril fusion and gluing strategy, the authors determined the cryo-EM structures of H1R bound to three inverse agonists, mepyramine, astemizole and desloratadine, as well as H1R in apo state. Structural analysis uncovered another extended ligand binding pocket in H1R when occupied by astemizole, and revealed the structural determinants about how these three agonists exert inverse agonistic activity. Overall, this is a nice work which provides new insights into the ligand binding and inverse agonism of H1R by antihistamines. However, the quality of this paper is restricted by the lack of pharmacological profiling and the limitation of the methodology regarding to the structure determination. My comments:

We appreciate very much that Reviewer #2 thinks this is a nice work that provides new insights into understanding of H1R with antihistamines. For the mentioned limitations, we have conducted biochemical assays to verify our conclusions.

Major concerns:

1. The authors showed the recognition details of mepyramine, astemizole and desloratadine in H1R. In the main orthosteric pocket of H1R, although these ligands share a similar set of interaction residues, differences in the residue composition and conformation could be observed, which may account for their distinct ligand potency. The authors also declared that the proper interaction of the ligand with extended pocket of H1R occupied by methoxyphenyl moiety of astemizole is critical for the ligand affinity and specificity. However, this point is also not supported by any pharmacological data. Would mutation of the polar residues K179, Y87 or N84 in the extended pocket into hydrophobic residues increases the potency of astemizole to H1R? Additional mutational assays using either IP1 accumulation or radioligand binding would be encouraged to verify the ligand poses and further clarify the detailed structural determinants in the main orthosteric pocket and the extended pocket for ligand potency.

We have performed functional assays by using IP1 accumulation to confirm the importance of the secondary pocket.

1. As shown in the revised Fig. 2c (also attached below), mutation of Y87 or W103, two residues from the secondary pocket that form critical hydrophobic interactions with the methoxyphenyl moiety of astemizole, remarkably decreases the potency/efficacy of astemizole. These results demonstrate that the hydrophobic contacts in the secondary pocket indeed contribute to the binding of astemizole to H₁R.

2. Accordingly, as described in the manuscript, the polar residue H450^{7.35} from the secondary pocket does not create a friendly environment for the methoxyphenyl moiety of astemizole; therefore, mutating H450^{7.35} to alanine improves the efficacy of astemizole (Fig. 2c). In contrast, in the mizolastine-H₁R docking model, H450^{7.35} mediates a hydrogen bond with the dihydropyrimidine moiety of mizolastine, and its mutation indeed cripples the potency of mizolastine (Fig. 2e). Taken together, these results further validate the concept that the interaction of antihistamines with the H₁R secondary pocket deserves optimization to improve potency and efficacy.

3. Another polar residue from the secondary pocket is N84. However, mutating N84 to alanine or valine did not improve the potency or efficacy of astemizole (see below). We speculate that because N84 is located in a deeper position in the pocket and has more contacts with surrounding residues such as W455 and Y458 (see below), its mutation may cause unexpected effects on the entire pocket or environment. Nevertheless, based on the structural observation and above functional results, we believe the conclusion of the manuscript is plausible.

2. H1R has high basal activity. It is interesting to see that the apo H1R structure reported in this study is in its inactive conformation, as the TM6 inwardly moved. It well known in GPCR area that the outward kink of TM6 is critical feature of GPCR activation, which is necessary for the opening of the receptor intracellular cavity for G protein coupling. As the inactive apo H1R conformation is not favorable for G protein coupling, one would think this inactive apo conformation maybe caused by the tight tethering of the mBril fusion and the Helix8 of H1R, rather than the real apo state of the receptor. Thus, the authors should tune down the discussion on the mechanism of basal activity of H1R and add more discussion on the limitation of the methodology in determination of apo GPCR structure.

We agree with reviewer #2. However, as described in our previous paper (<https://doi.org/10.1038/s41589-023-01389-0>), in our experience, the mBril fusion and gluing strategy is unable to fix the conformation of TM6. For example, the active-like conformations of TM6 and TM7 were captured in β 2 receptors bound to partial agonists. The mBril fusion and bonding strategy only fixes the conformation of mBril but not that of TM6 (shown below).

3. All the three ligands, mepyramine, astemizole and desloratadine, behave as selective H1R inverse agonists. More structural comparison of these inverse agonists bound H1R structures with other histamine receptor subtypes (H2R, H3R and H4R) would further elucidate the ligand selectivity determinants of histamine receptor family.

Thanks to Reviewer #2. We have added a detailed structural comparison in Supplementary Fig. 10. We also added more discussion to the manuscript on ligand selectivity.

Minor issues:

1. Line 161: change 'orthosteric pocket mainly constituted by TM3...' into 'orthosteric pocket mainly constituted by residues from TM3...'

Thanks a lot.

2. Line 181: 'As anticipated' should be 'as anticipated'

Thanks a lot.

3. Line 221: change 'defined by ECL2...' into 'defined by residues from ECL2...'

Thanks a lot.

4. Line 230: 'promises to' is not properly used in this sentence, please correct it.

Thanks a lot. We have made the corrections.

5. Some H1R ligands such as oxatomide were reported to be neutral antagonists of H1R, are there any differences in the ligand binding between H1R neutral antagonist and inverse agonist? Perhaps the authors could do some molecular docking work to gain more insights into the ligand binding property of H1R.

We read the original paper that classified oxatomide into antagonist (<https://doi.org/10.1016/j.intimp.2013.02.009>), however, in that paper, they also define loratadine as an antagonist. However, according to other studies (<https://doi.org/10.1159/000082325>, [https://doi.org/10.1016/s0014-2999\(99\)00803-1](https://doi.org/10.1016/s0014-2999(99)00803-1)), loratadine acts as an inverse agonist instead of antagonist. Additionally, based on our assays, loratadine is also an inverse agonist. As a result, we do not think that the claim that oxatomide is an antagonist is true. Furthermore, according to these papers (<https://doi.org/10.1046/j.0954-7894.2002.01314.x>, <https://doi.org/10.4103/0019-5154.110832>, <https://doi.org/10.1111/exd.14602>, <https://doi.org/10.1016/B978-0-7020-7167-6.00039-7>), H1-antihistamines are all inverse agonists rather than antagonists. However, to be more accurate, we have changed the description of "all H1R blockers act as inverse agonists" in the original manuscript to "all validated H1R blockers act as inverse agonists".

Reviewer #3 (Remarks to the Author):

In this study, Wang, et al. present four cryo-EM structures of H1R in apo and antihistamine-bound states. These structures show the H1R binding details of three inverse agonists, including first-generation antihistamine mepyramine and second-generation ones astemizole and desloratadine. They also identify a secondary ligand-binding site that may support the introduction of new derivative groups to generate newer antihistamines. The activation mechanism of H1R is also been discussed. This work is of importance in understanding antihistamine recognition by H1R and guiding new-generation antihistamines targeting H1R. However, there are still issues to be addressed.

We thank Reviewer #3 for the comments on our work. We appreciate very much that Reviewer #3 thinks our work is important.

Major concerns:

1. The clash scores for the mepyramine- and desloratadine-bound H1R structures exceed 36 and 47, respectively. It is strongly recommended to further optimize these models to obtain more reliable and convincing models. Additionally, there seems to be an inversion in the data for favored (%) and allowed (%), which should be rectified.

We have refined the structural models. We have also corrected the number of favored (%) and allowed (%).

2. -Lines 100-116, since reference 28 has not been published, the audiences may be confused by the gluing strategy and further the design of "4-9" and "6-13" glues. To ensure clarity, it is essential to provide the necessary information to elucidate the gluing strategy and the design of these specific glues.

We have updated reference 28. Now, it should be easy to understand the method.

3. -Lines 130 and 131, the statement that "without inverse agonist stabilization, H1R may have a dynamic structure, reminiscent of its high basal signaling capacity" is incorrect. In the apo state, most GPCRs exist in an equilibrium between different conformational states. This equilibrium is why only a few GPCR structures have been solved without ligands or downstream signaling proteins. Therefore, it is not convincing to correlate the lower density maps and dynamic conformation of apo H1R with its "high basal signaling capacity."

Thanks. We have modified the description.

4. -Lines 228-246, the author's assertion regarding 'modifying the methoxyphenyl moiety with alternative derivative groups, for example, introducing hydrogen bond donors/acceptors, may further improve the affinity and specificity of astemizole' is not well supported by the subsequent description. It is important to address whether there are any differences in ligand affinity between mizolastine and astemizole. Additionally, it would be valuable to discuss whether the sequence differences at the secondary pocket across the four histamine subtypes support the high selectivity of mizolastine over H1R. Further discussions on these issues would provide stronger support for the author's hypothesis.

We have changed the description in the revised manuscript. For this section, we meant to show that full compatibility between the derivative group and the secondary pocket could increase the affinity and/or specificity of the ligand. For mizolastine and astemizole, we were not comparing their affinity differences. However, as shown in the revised Fig. 2c, we can see that the hydrophobic interactions between astemizole and the secondary pocket (mediated by Y87 and W103) contribute to the inhibition capacity of astemizole, and the hydrogen bond between mizolastine and the secondary pocket (mediated by H450^{7,35}) also contributes to the inhibition capacity of mizolastine (Fig. 2e). Therefore, it is conceivable that it would be more reasonable to use a combination of hydrophobic and hydrophilic interactions between the ligand and the pocket. As a result, further optimization of the derivative group, for example, designing a derivative group that maintains hydrophobic interactions but can also form a hydrogen bond with the secondary pocket, may yield antihistamines with improved affinity/potency/efficacy.

Finally, as suggested, we have included more discussions (also Supplementary Fig. 10, 11) in the revised manuscript on the residues around the main and secondary pocket,

highlighting the sequence divergence on the secondary pocket that may be useful for the development of new antihistamines with improved specificity/affinity.

5. Given that the identification of the second binding pocket and its potential contribution to subtype selectivity constitutes a crucial aspect of this study, it is imperative to conduct mutagenesis assays and other functional analyses, rather than relying on molecular docking, to provide robust evidence supporting the author's hypothesis.

We have performed functional assays to confirm the conclusion of this work.

1. As described in the manuscript, we found that the methoxyphenyl moiety of astemizole forms hydrophobic interactions with residues in the secondary pocket. Signaling assays showed that disruption of these interactions indeed impairs the ability of astemizole to inhibit H₁R signaling (Fig. 2c). In addition, the polar residue H450^{7,35} from the secondary pocket is incompatible with the hydrophobic nature of the methoxyphenyl moiety, and as expected, the H450A mutation improves the efficacy of astemizole (Fig. 2c). In contrast, in the mizolastine-H₁R model, H450^{7,35} forms a hydrogen bond with the dihydropyrimidine of mizolastine, and the H450A mutation indeed cripples the inhibition ability of mizolastine (unlike the boosting effect for astemizole) (Fig. 2e).

2. For loratadine, as described in the manuscript, its derivative group in the secondary pocket cannot form effective hydrophobic interactions with the hydrophobic residues in the secondary pocket. Instead, the carbonyl tail even faces two hydrophilic atoms from H₁R residues (N84 and Y458). Therefore, its potency in inhibiting H₁R is significantly lower than that of desloratadine (Fig. 2h).

Taken together, all the results of functional assays support our conclusion.

6. -Fig. 2c,e, it appears that mizolastine and loratadine, in the docked models, penetrate deeper into the binding site compared to astemizole and desloratadine in the experimental models, respectively. For instance, D1073.32 establishes an electrostatic interaction with the N atom in the piperidine ring of desloratadine, whereas it interacts with the O atom in the carboxylate group of loratadine positioned above the piperidine ring. Has the author attempted to merge the experimental and docked models to compare the experimental and docked models? It would be intriguing to explore whether they exhibit different binding depths despite sharing the same tricyclic chemical group.

The view of Fig. 2c and Fig. 2e (Now as Supplementary Fig. 2d and Fig. 2g in the revised version) probably results in the impression of Reviewer #3. Actually, the binding depths of loratadine and desloratadine are very similar in the ligand-binding pocket. As shown below, the ligand binding poses superimpose well after aligning the experimental and docked models.

The binding depths of mizolastine and astemizole are also very similar (see below).

7. -Lines 326-329, the author should be very careful to conclude 'it adopts a typical inactive conformation in the apo state'. The fusion protein attaching to ICL3 may modify the equilibria between different conformational states (Eddy, M. T., et al. Structure 24, 2190–2197, 2016), especially the two rigid linkers between TM5, TM6, and N-, C-term of BRIL, respectively. That is, the glue strategy may strongly restrict and force H1R to the inactive state to leading false judgment.

We agree with reviewer #3. However, as described in our previous paper (<https://doi.org/10.1038/s41589-023-01389-0>), in our experience, the mBril fusion and gluing strategy is unable to fix the conformation of TM6. For example, the active-like conformations of TM6 and TM7 were captured in β_2 receptors bound to partial agonists. The mBril fusion and bonding strategy fixes only the conformation of mBril and not that of TM6 (shown below).

8. -Lines 320-350, the reviewer noticed that a 'squash to activate and expand to deactivate' model has been proposed for H₁R previously (Xia, et al., Nat Commun. 2021, 12(1):2086). According to this model, the salt-bridge interaction between TM3 and TM6 is suggested to mediate the basal activity of H₁R, while the bulky tricyclic groups of inverse agonists induce an inactive conformation by pushing phenylalanine in TM6. The question arises whether the author's findings align with these previous conclusions or provide new insights into H₁R activation. It is important to include a discussion addressing this point and discussing the consistency or novelty of the author's findings concerning the previously proposed model.

In general, our results support the 'squash to activate and expand to deactivate' model. By comparing the structures of inverse agonist-, agonist-bound and apo structures, we found that the inverse agonist-bound structure has the widest opening of TM6 on the extracellular side and that the histamine-bound structure has the most contracted structure. In the apo state, TM6 and TM7 appear to adopt an intermediate conformation (Supplementary Fig. 8). The readily close movement between TM3 and TM6 in the apo state may support the basal signaling ability of H₁R. We have added these discussions to the revised manuscript.

9. -Lines 375-377, reference 52 is incorrect. Please ensure the correct reference is cited. Additionally, based on information from the IUPHAR database (<https://www.guidetopharmacology.org/>), it is stated that blockers for all four histamine receptor subtypes function as both inverse agonists and neutral antagonists. It is recommended to double-check this information for accuracy.

We have corrected the relevant literature. The information from the IUPHARA database is not the latest results. The IUPHAR database reports that some H₁R blockers, such as desloratadine and loratadine, are antagonists; however, according to the literature (<https://doi.org/10.1159/000082325>) and ([https://doi.org/10.1016/s0014-2999\(99\)00803-1](https://doi.org/10.1016/s0014-2999(99)00803-1)), both desloratadine and loratadine act as inverse agonists, not agonists. Consistently, our functional assays have also demonstrated that they function as inverse agonists. Furthermore, according to these papers (<https://doi.org/10.1046/j.0954-7894.2002.01314.x>, <https://doi.org/10.4103/0019-5154.110832>,

<https://doi.org/10.1111/exd.14602>, <https://doi.org/10.1016/B978-0-7020-7167-6.00039-7>), H1-antihistamines are all inverse agonists rather than antagonists. However, to be more accurate, we have changed the description of “all H1R blockers act as inverse agonists” in the original manuscript to “all validated H1R blockers act as inverse agonists”.

10. -Lines 382-390, to the reviewer’s knowledge, there is no evidence supporting the claim that the direct interaction between blockers and the toggle switch W6.48 can distinguish inverse agonists from neutral antagonists. Please cite corresponding references to support the author’s statement.

For this section, we meant to claim that owing to having different chemical structures and ligand binding pockets (with respect to H₁R ligands and H₁R ligand pocket), the ligands for other histamine receptors (H₂R, H₃R and H₄R) employ a different modulation mechanism, unlike the inverse modulation mechanism of H₁R antihistamines. To be more accurate, we have changed the description in the revised manuscript. However, to be more accurate, we have changed the description of “all H1R blockers act as inverse agonists” in the original manuscript to “all validated H1R blockers act as inverse agonists”.

Minor concerns:

1. -Fig.1, the EM density of several helices of H1R is shown inconinuous. Please adjust the EM threshold. In addition, the names of ligands are suggested to be labeled and the sticks of ligands should be colored by heteroatom.

We have adjusted the EM threshold for the density map and added the ligand names. As suggested, the ligands are now colored by heteroatoms.

2. -Fig.2a, W4326.52 should be F4326.52.

Thanks a lot.

3. -Fig.2a,e, W1083.33 should be Y1083.33.

Thank you, we have made the corrections in Fig. 2a and Fig. 2e (now Fig. 2a and 2g in the revised version).

4. -Fig. S8, H3R-PF-03654746 (PDB ID: 7D6L) is incorrect, which should be 7F61.

Thanks a lot, we have made correction to the errors.

5. -Line 427, sf9 insect cells at a density of 4 x 10³ per ml are unreasonable, please double-check.

Thanks a lot, we have corrected the previous errors.

6. -Line 463, ‘complexex’ should be ‘complexes’.

Thanks a lot, we have made corrections to the errors.

REVIEWERS' COMMENTS

Reviewer #1 (Remarks to the Author):

My concerns are fully addressed

Reviewer #2 (Remarks to the Author):

The authors have provided additional data and proper discussion to address most of my concern. Still, I have a few questions regarding the pharmacological data and the conclusion the authors made.

1. The authors only did mutational assays on the critical residues in the secondary pocket. However, the structural determinants on ligand potency in the main pocket, which are also important for the rational design of new antihistamines, were not elucidated. Mepyramine and desloratadine share conserved binding pose but differ in ligand potency, which could be caused by divergent interaction with certain residues in the main pocket. I want to see more data and discussion on the influence of the main pocket on the potency of different ligands used in this study, especially mepyramine and desloratadine.

2. What effect will mutating K179 into hydrophobic residues have on the potency of astemizole to H1R? This point should be clarified.

3. Please list all the EC50 and Emax data of IP1 accumulation assay on H1R WT and mutants in table form.

Reviewer #3 (Remarks to the Author):

All Reviewer's concerns have been addressed.

Other Concerns:

The statistics of all mutagenesis data, including IC50 and Emax values, is suggested to be summarized in a supplementary table.

Reviewer #1

(Remarks to the Author):

My concerns are fully addressed

Reviewer #2

(Remarks to the Author):

The authors have provided additional data and proper discussion to address most of my concern. Still, I have a few questions regarding the pharmacological data and the conclusion the authors made.

1. The authors only did mutational assays on the critical residues in the secondary pocket. However, the structural determinants on ligand potency in the main pocket, which are also important for the rational design of new antihistamines, were not elucidated. Mepyramine and desloratadine share conserved binding pose but differ in ligand potency, which could be caused by divergent interaction with certain residues in the main pocket. I want to see more data and discussion on the influence of the main pocket on the potency of different ligands used in this study, especially mepyramine and desloratadine.

We have updated the manuscript and included more discussions on the main pocket residues.

1. In fact, many groups before have confirmed the indispensability of the main pocket residues. As described in the "Recognition mechanism of mepyramine with H₁R" section of the manuscript, we mapped the previous mutagenesis results onto the structure of mepyramine-H₁R. Based on our structure and previous biochemical results, the importance of the main pocket residues has already been elucidated.

2. As suggested, we have also performed additional signaling assays with H₁R mutants focusing on main pocket. Specially, two residues Y431^{6.51} and F432^{6.52} from the main pocket, which make hydrophobic interactions with the core groups of mepyramine and desloratadine, were mutated to alanine for signaling assays (in accordance with previously reported results, mutations in most major pocket residues completely disrupted the signaling ability of H₁R, thus hampering signaling assays. However, both Y431A and F432A mutants have basal signaling capabilities, so we selected these two mutants). As shown in Supplementary Fig. 8 (also below), both mutations cause mepyramine and desloratadine to lose their ability to inhibit H₁R signaling, again demonstrating the importance of major pocket residues.

3. Finally, most antihistamines including mepyramine and desloratadine here have different chemical compositions (as shown below) and their combinatorial interactions with H₁R pocket residues result in ligands with varying potencies. Therefore, it's difficult to elucidate the contribution of each residue to ligand potency. However, as mentioned above, we can conclude that the residues from the main pocket are indispensable for the ligand to execute inhibitory functions.

2. What effect will mutating K179 into hydrophobic residues have on the potency of astemizole to H₁R? This point should be clarified.

We have performed the signaling assays on K179 mutants. Since K179 is located further outside and slightly away from the methoxyphenyl moiety of astemizole, we mutated K179 to tyrosine and phenylalanine, respectively, two hydrophobic residues with bulky side chains. As shown below, while K179F did not improve the efficacy of astemizole, K179Y induced a better efficacy. This result again supports the conclusion of the manuscript.

3. Please list all the EC50 and Emax data of IP1 accumulation assay on H1R WT and mutants in table form.

We have added the IC50 and Emax data on H1R WT and mutants in Supplementary Table 2.

Reviewer #3

(Remarks to the Author):

All Reviewer's concerns have been addressed.

Thanks a lot to the Reviewer #3.

Other Concerns:

The statistics of all mutagenesis data, including IC50 and Emax values, is suggested to be summarized in a supplementary table.

We have added the statistics of all mutagenesis data, including IC50 and Emax value in Supplementary Table 2.